

# Solving the inverse heat conduction problem using NVLink capable Power architecture

Sándor Szénási

John von Neumann Faculty of Informatics, Óbuda University, Budapest, Hungary

## ABSTRACT

The accurate knowledge of Heat Transfer Coefficients is essential for the design of precise heat transfer operations. The determination of these values requires Inverse Heat Transfer Calculations, which are usually based on heuristic optimisation techniques, like Genetic Algorithms or Particle Swarm Optimisation. The main bottleneck of these heuristics is the high computational demand of the cost function calculation, which is usually based on heat transfer simulations producing the thermal history of the workpiece at given locations. This Direct Heat Transfer Calculation is a well parallelisable process, making it feasible to implement an efficient GPU kernel for this purpose. This paper presents a novel step forward: based on the special requirements of the heuristics solving the inverse problem (executing hundreds of simulations in a parallel fashion at the end of each iteration), it is possible to gain a higher level of parallelism using multiple graphics accelerators. The results show that this implementation (running on 4 GPUs) is about 120 times faster than a traditional CPU implementation using 20 cores. The latest developments of the GPU-based High Power Computations area were also analysed, like the new NVLink connection between the host and the devices, which tries to solve the long time existing data transfer handicap of GPU programming.

# INTRODUCTION

As a fundamental experience of modern materials science, material properties are influenced by the microstructure; therefore, these can be altered to improve the mechanical attributes (*Oksman et al., 2014*). One of the most widely used methods for this purpose is heat treatment which usually consists of two consecutive steps: heating up the work object to a given high temperature and cooling it down in a precisely controlled environment. It is necessary to know the attributes of the given material and the environment, to achieve the best results, especially the Heat Transfer Coefficient (HTC) which shows the amount of heat exchanged between the object and the surrounding cooling medium.

The Inverse Heat Conduction Problem (IHCP—the determination of the HTC) is a typical ill-posed problem (*Beck, Blackwell & Clair st, 1985*; *Alifanov, 1994*; *Felde, 2016b*). Without any known analytical solution, most methods are based on the comparison

Corresponding author
Sándor Szénási,
szenasi.sandor@nik.uni-obuda.hu

of temperature signals recorded during real heat treatment processes and estimated by simulations. The aim of these methods is to find the HTC function giving the minimal deviation of the measured and predicted temperature data.

It is usual to use heuristic algorithms, like Genetic Algorithms (GAs) (*Szénási & Felde, 2017*), Particle Swarm Optimisation (PSO) (*Felde & Szénási, 2016*; *Szénási & Felde, 2015*) or some hybrid approaches (*Felde, 2016a*) to find this parameter. *Kim & Baek (2004)* presented a hybrid genetic algorithm for the analysis of inverse surface radiation in an axisymmetric cylindrical enclosure. *Verma & Balaji (2007)* used a stochastic Particle Swarm Optimization to estimate several parameters in the field of inverse conduction-radiation. Both papers have significant contribution in the field of inverse methods.

In the case of GAs, every chromosome of the population encodes one possible HTC function in its genes. These are two-dimensional continuous functions given by the time and location. Therefore, a limited number of control points have been used to approximate these (340 floating point values per HTC). With the already existing Direct Heat Conduction Problem (DHCP) solver methods (based on finite-elements or finite-difference techniques), it is feasible to simulate the cooling process and to record the thermal history for each chromosome. The difference between this generated thermal history and the measured one produces the cost value for the individual. The purpose of the IHCP process is to find the best gene values resulting in minimal cost.

The bottleneck of this process is the high computational demand. The runtime of one cooling process simulation is about 200 ms using one traditional CPU core, and it is necessary to run these simulations for each chromosome in each iteration. Assuming a population of 2,000 chromosomes and a GA with 3,000 iterations, it takes several days to finish the search. Furthermore, according to the random behaviour of the GA and the enormously large search space, it is worth running multiple searches. As a result, an overall IHCP process can take many weeks.

There are several attempts at using graphics accelerators to speed up physical simulations, and there are several substantial achievements in this field too. *Satake, Yoshimori & Suzuki (2012)* presented a related method to solve heat conduction equations using the CUDA Fortran language. They worked out a very well optimised method (analysing the PTX code generated by the compiler), but they only dealt with the one-dimensional unsteady heat conduction equation for temperature distribution.

*Klimeš & Štětina (2015)* presented another model using the finite difference method to simulate the three-dimensional heat transfer. Their results showed that the GPU implementation is 33–68 times faster than the same CPU-based simulation using one Tesla C2075 GPU for running kernels. This significant speed up makes it possible to use their method in a real-time fashion. *Narang, Wu & Shakur (2012)* and *Narang, Wu & Shakur (2016)* also used programmable graphics hardware to speed up the heat transfer calculations based on a similar finite difference method. They used a Quadro FX 4800 card, and the speed up was still significant (about 20× in the case of a large number of nodes).

There are also several papers from similar areas. *Humphrey et al. (2012)* and *He et al. (2013)* have very significant contributions to the field of radiative heat transfer problems. These also show that it is worth implementing GPU codes for physical simulations.

The main difference between these studies and this research is that this paper is focusing on the two-dimensional IHCP. Heat transfer simulation is a major part of the IHCP solving process; moreover, it is necessary to run thousands of simulations. Accordingly, it is feasible to use a higher level of parallelism by using multi-GPU architectures (the presented papers are usually deal with only one device). It is possible to install multiple graphics cards into a standard PC motherboard, and the CUDA programming environment can handle all of them. Using multiple GPUs can double/triple/quadruple the processing power, but it is necessary to adapt the algorithms to this higher level of parallelism.

One of the most interesting developments of 2016 in HTC computing is the result of the IBM and NVIDIA collaboration, the NVLink high-speed interface between IBM's POWER8 CPUs and NVIDIA's Pascal GPUs. Data transfer between the host and device memory was an important bottleneck in GPU programming, making several promising applications practically unusable. This high-speed connection and the existence of multiple Pascal based graphics cards give developers the ability to accelerate applications more easily. The assumption was that it is worth implementing an IHCP solver system based on this architecture.

Based on these advancements, a novel numerical approach and a massively parallel implementation to estimate the theoretical thermal history are outlined. The rest of the paper is structured as follows: the next section presents the novel parallel DHCP and IHCP solver methods; 'Results and Discussion' presents the raw results of the benchmarks and the detailed analysis; finally, the last section contains the conclusions and further plans.

## MATERIALS & METHODS

### Direct heat conduction problem

There are various fundamental modes of heat transfer, but this paper deals only with transient conduction when the temperature of the workpiece changes as a function of time. The determination of the temperature of any points of the object at any moment often calls for some computer-assisted approximation methods. In the case of three-dimensional objects, these calculations can be very complicated and resource intensive, preventing the efficient usage as a GA cost function. For cylindrical objects, an essential simplification can significantly decrease the computational effort. As can be seen in Fig. 1 it is enough to model the middle cross-section of the cylinder, resulting in the two-dimensional axis-symmetrical heat conduction model. The radius of the cylinder is noted by R and Z.

The cylinder is subjected to a longitudinal local coordinate and time varying Heat Transfer Coefficient $HTC(z, t)$ on all its surfaces. Both the thermal conductivity, density and the heat capacity are varying with the temperature, $k(T)$, $\rho(T)$ and $C_p(T)$. It has to be noted that phase transformations of the materials applied do not occur during the experiments, therefore latent heat generation induced by phase transformations is not considered.

Based on this simplified model, the mathematical formulation of the nonlinear transient heat conduction can be described as Eq. (1), with the following initial and boundary

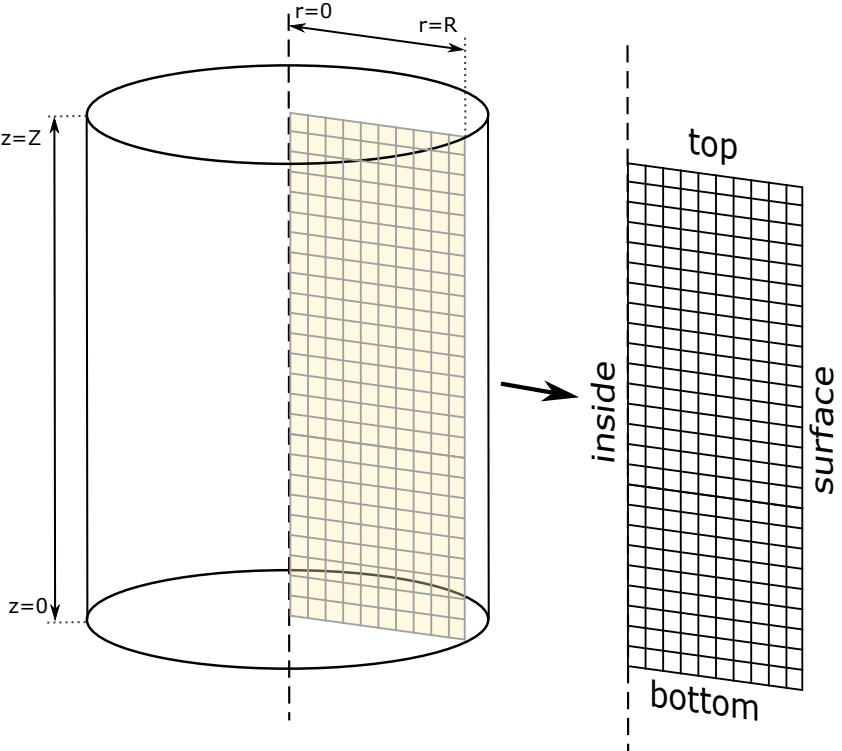

**Figure 1** Two-dimensional axis-symmetrical heat conduction model.

conditions Eqs. (2)–(5):

$$\frac{\partial}{\partial r}(k\frac{\partial T}{\partial r}) + \frac{k}{r}\frac{\partial T}{\partial r} + \frac{\partial}{\partial z}(k\frac{\partial T}{\partial z}) + q_v = \rho C_p \frac{\partial T}{\partial t} \tag{1}$$

$$T(r=0, z, t=0) = T_0, \tag{2}$$

$$k\frac{\partial T}{\partial z}|_{0 \leq z \leq L, r=R} = HTC(z,t)[T_q - T(r=R, z, t)], \tag{3}$$

$$k\frac{\partial T}{\partial z}|_{z=0, 0 \leq r < R} = HTC(z=0,t)[T_q - T(r, z=0, t)], \tag{4}$$

$$k\frac{\partial T}{\partial z}|_{z=Z, 0 \leq r < R} = HTC(z=Z,t)[T_q - T(r, z=Z, t)], \tag{5}$$

where

- $r, z$—local coordinates;
- $t$—time;
- $R$—radius of the workpiece;
- $\rho(T)$—density of the object;
- $T_0$—initial temperature of the workpiece;
- $T_q$—temperature of the cooling medium;
- $T(r, z, t)$—temperature of the workpiece at given location/time;
- $k(T)$—thermal conductivity (varying with temperature);

- $C_p(T)$—heat capacity (varying with temperature);
- $HTC(z, t)$—heat conduction (varying with local coordinate and time).

Discretising of these equations using the weighted Schmidt explicit finite difference method, results in nine different equations, according to the location within the object.

- case (a)—inner items surrounded by four neighbouring items;
- case (b)—inner items at the centre line of the object;
- case (c)—boundary items at the outer surface of the object;
- case (d)—boundary items at the top surface of the object;
- case (e)—boundary items at the bottom surface of the object;
- case (f)—boundary item at the outer top corner;
- case (g)—boundary item at the inner top corner;
- case (h)—boundary item at the outer bottom corner;
- case (i)—boundary item at the inner bottom corner.

The reliability of the estimated HTC strongly depends on the numerical approach applied. During the DHCP calculations, the resolution of the finite grid is essential, parameters $n = 10$, $m = 34$ were used, where $n$ is the number of points horizontally, and $m$ is the number of points vertically. A sufficiently small $dt$ (simulation time interval) value is also necessary to ensure the accuracy of the method ($dt = 0.01$ second).

## Massively parallel solution

It is necessary to solve the heat transfer equations for each finite item for the consequent time periods. Equations for calculating the heat movement between neighbouring items within the same time can be solved in a parallel fashion because none of these calculations modifies the output of the others. Moreover, for several items, the steps of these calculations are the same, only the input parameters are different. This makes it possible to implement an efficient data-parallel algorithm. Assuming $n \times m$ number of threads, the parallelization level is:

- $(n - 2) \times (m - 2)$ threads have to calculate case a;
- $(m - 2)$ threads have to calculate case b;
- $(m - 2)$ threads have to calculate case c;
- $(n - 2)$ threads have to calculate case d;
- $(n - 2)$ threads have to calculate case e;
- $1 + 1 + 1 + 1$ threads have to calculate case f, g, h and i.

Based on this data parallel fashion, it is worth implementing a GPU-based implementation. Every GPU thread is responsible for one item in the finite grid. The task of these threads is the determination of the temperature in the corresponding location in each consecutive time step. These threads can work in parallel with the equations of the same time step. Nevertheless, these are not totally independent, because as an input parameter, threads need the temperature of the neighbouring elements at the previous time step. The threads have to wait until all the others have finished the calculations for the actual period to fulfil this before they can continue working on the next time step.

This needs an explicit synchronisation after every time step, and in the CUDA framework, the only efficient barrier synchronisation is given by the __*syncthreads()* function, which synchronises the threads within the same block. According to this, it is necessary to keep all threads working on the same finite grid in the same CUDA block. There is an upper limit for the number of threads within a block (1,024 for current architectures); but, in the case of ordinary configuration, this is not a limiting factor ($10 \cdot 34 = 340$ threads were used).

Running 340 threads is not sufficient to fully utilise a modern GPU. One P100 accelerator card has 3,584 CUDA cores. Therefore, this low number of threads leads only a low theoretical occupancy level (number of used cores/number of available cores = 340/3,584 = 9.48%), and the practical utilisation is expected to be even worse (not to mention, that one server node has four cards installed). At this point, recall that this DHCP calculation is responsible for the cost calculation part of a genetic algorithm. A population containing $P$ chromosomes needs $P$ number of thermal history generations in the evaluation phase of the GA. These are independent calculations, so it is feasible to launch these in a parallel fashion using more than one multi-processors. Given this higher level of parallelism, the required number of threads becomes $P \times 340$, which is enough to utilise the computing power of the graphics accelerators fully.

This observation is the key point for the multi-GPU implementation. The already mentioned independence makes it possible to distribute these fitness calculations among the available devices. In the case of $G$ number of GPUs, it is worth to assign each chromosome to one of the GPUs using Eq. (6), which shows the number of assigned chromosomes ($P_i$) for the $i$th GPU (where $1 \leq i \leq G$).

$$P_i = \begin{cases} P - (G-1) \left\lfloor \dfrac{P}{G} \right\rfloor, & \text{if } i = 1 \\ \left\lfloor \dfrac{P}{G} \right\rfloor, & \text{otherwise} \end{cases} \tag{6}$$

## Further optimisation
### Using shared memory
GPU applications can easily be limited by memory bandwidth issues. Data starvation occurs when all threads of a block must wait for loading or storing the actual temperature value of the corresponding finite item. Taking the advantages of the heterogeneous memory hierarchy makes it feasible to decrease this adverse effect. Chromosome data (the HTC functions) and the fitness values must be in the device memory of the GPU because these must be transferred from and to the host. However, during the thermal history generation, it is better to store the actual temperature data of the finite grid in the fast on-chip shared memory. All threads of the block can read and modify these values. Therefore, these can read the actual temperature values of the neighbouring elements.

The adverse effect of heavy shared memory usage is the limit for the number of parallel block executions. The P100 architecture has 64 KB of shared memory per multiprocessors. 340 float variables are necessary to store the actual values of the grid, requesting only 1,360 bytes of shared memory by simulations. According to this, even 48 blocks can run in parallel

| g | d | d | d | d | d | d | d | d | f |
|---|---|---|---|---|---|---|---|---|---|
| b | a | a | a | a | a | a | a | a | c |
| b | a | a | a | a | a | a | a | a | c |
| b | a |   |   |   |   |   |   |   |   |

**Figure 2 Heavy warp divergence at the top of the workpiece.** Letters show the locations linked to threads in the first warp. Different letters represent different code paths.

in one multiprocessor. Other constraints (maximum number of resident warps/threads by multiprocessor) have much stronger limits, and thus shared memory usage does not cause any disadvantage. The implementation of this on-chip memory usage gives about 2–3× speed-up for the GPU implementation (based on other parameters, like population size, grid size, full simulation time).

### Warp divergence

When a thread block is assigned to a multiprocessor, it is divided into further groups of 32 threads called warps. This partitioning is predetermined: threads with index 0–31 will be the first warp, threads in the 32–63 index range the second warp, and so on *NVIDIA (2014)*. The warp is the lowest unit of execution scheduling, at any time, one instruction is fetched and executed by all threads within the same warp. The operands of these instructions can be different; therefore, in the case of conditional statements, different threads in the same warp should take different paths. The solution for this warp divergence is that all threads within a warp must take both branches of the conditional statement; threads with false condition are stalling, while the others are executing the instructions of the true branch, and threads with true condition are stalling, while the others are executing the false branch. These stall phases can significantly decrease the occupancy; hence, warp divergence should be avoided to obtain the best performance.

In the case of DHCP calculations, the number of threads is equal to the size of the finite grid ($n \times m$). Every thread corresponds to the calculations of one item in the grid, and the first intuition shows that thread indices should be the same as the indices of the corresponding finite item. However, this leads to heavy warp divergence; for example, threads of the first warp (identified by indices below) should execute the following code paths (Fig. 2):

- (0, 0) thread have to calculate case g;
- (9, 0) thread have to calculate case f;
- (1, 0)–(8, 0) threads have to calculate case d;
- (0, 1)–(0, 3) threads have to calculate case b;
- (9, 1)–(9, 2) threads have to calculate case c;
- (1, 1)–(8, 2) and (1, 3) threads have to calculate case a.

Separation of thread indices and the corresponding finite grid locations makes it possible to decrease this divergence significantly. Threads inside the first warp should calculate the equations corresponding to the elements at the centre line (32 items). The second warp should correspond to the outer surface elements (32 items). Threads in the next eight warps should be responsible for the inner elements (272 elements), and the last warp should solve all the remaining equations (20 elements).

In this partitioning, only the last warp has divergent threads. The tests show that the gained speed-up is about 10–20%.

### Algorithm description

The DHCP function of Algorithm 1 contains the main host-side steps of the GPU based Direct Heat Conduction Solver algorithm including device memory allocation/deallocation, memory transfers to/from the device and the kernel launch. The DHCP-Kernel function shows the device-side steps of the heat transfer simulation according to the presented optimisation steps.

## RESULTS AND DISCUSSION

### Benchmarking methodology

Several benchmarks were run with the CPU and the GPU implementations focusing on the following questions:

- What is the correlation between the number of CPU cores/GPU devices and the required runtime? A linear correlation is expected because of the weak dependencies between the different tasks.
- In the case of GAs, which hardware configuration is preferred for a given population size? The expectation is that it is worth using the CPU for small populations and the GPU for larger populations.
- The amount of input parameters (HTC control points) and output results (fitness values) is relatively small compared to other HPC applications. Is the new NVLink technology between the CPU and GPU able to significantly reduce the memory transmission time for these small data transfers?

The details of the test environments are as follows:

- Test Environment 1 (TE1)
  - CPU—1 × Intel(R) Core(TM) i7-2600
    * 4 physical cores
    * 8 logical cores
  - GPU—2 × GeForce GTX TITAN Black
    * CUDA cores: 2880
    * Memory: 6 GB
    * Link: PCIe 4× + PCIe 8×

---

**Algorithm 1** Data parallel DHCP solver

---

**Require:** $HTC$: Heat Transfer Coefficient
**Require:** $T_0$: initial temperature
**Require:** $R$: reference point inside the work object
**Ensure:** $S[\ ]$: recorded temperature values at the reference point

1: **function** DHCP-KERNEL( )             ▷ threadId: unique identifier in $0...n \cdot m - 1$
2:      $(i, j) \leftarrow AssignThreadToFiniteNode(\mathbf{threadId})$           ▷ 'Warp divergence'
3:      $T_{i,j} \leftarrow T_0$     ▷ Shared array to store actual temp values ('Using shared memory')
4:      **synchronise threads**
5:      **for** $t \leftarrow 0$ *to S.length* $- 1$ **do**
6:          $time \leftarrow t * dt$                                 ▷ Actual time
7:          $temp \leftarrow T_{i,j}$             ▷ Thread-level variable (actual temp at (i,j) pos)
8:          **switch** $(i, j)$ **do**        ▷ Calculate heat movement ('Direct heat conduction problem')
9:              **case** $(0, 0)$
10:                 $temp \leftarrow temp + HeatTransferInnerTopCorner(T, HTC)$
11:              **case** …
12:                 …
13:          **end switch**
14:          **synchronise threads**
15:          $T_{i,j} \leftarrow temp$
16:          **synchronise threads**
17:          **if threadId** $= 0$ **then**            ▷ First thread calculates the result
18:              $S[t] \leftarrow InterpolateTempAt(T, R)$      ▷ Calculate temp at ref. point
19:          **end if**
20:      **end for**
21: **end function**
22:
23: **function** DHCP$(HTC, T_0, R)$
24:      ALLOCATEGPUMEMORY( )
25:      COPYFROMHOSTTODEVICE$(HTC, T_0, R)$
26:      DHCP-KERNEL $\lll n * m \ggg$ ( )           ▷ Launch $n \cdot m$ parallel threads
27:      COPYFROMDEVICETOHOST$(S)$
28:      FREEGPUMEMORY( )
29:      **return** $S$
30: **end function**

---

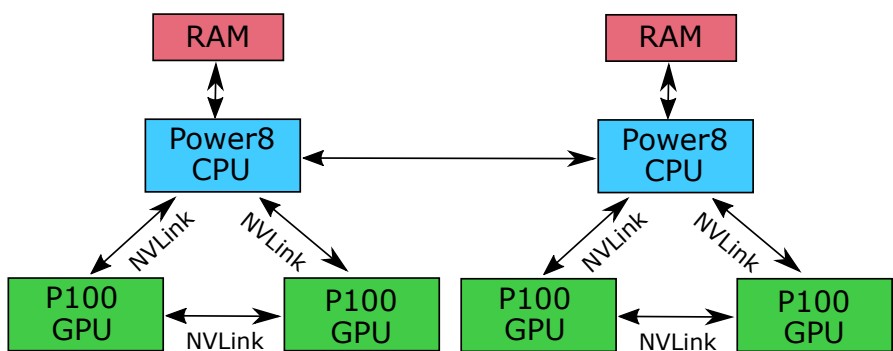

**Figure 3** **Architecture of the IBM Power System S822LC for High Performance Computing.**

- Test Environment 2 (TE1)
  - CPU—2 × IBM POWER8NVL
    * 10 cores
  - GPU—4 × Tesla P100-SXM2
    * CUDA cores: 3584
    * Memory: 16 GB
    * Link: PCIe 8×+ NVLink

TE1 was a Windows 10 based desktop machine with two GeForce GTX TITAN Black cards installed into the PCIe slots. The IHCP codebase was developed using Visual Studio 2015 (with CUDA 8.0) and compiled with the Nvidia nvcc compiler to standard 64 bit executables. These binaries were launched from the standard Windows command prompt.

The C ++ standard *std::chrono:high_resolution_clock* object was used to measure the execution time. To decrease uncertainty, 20 independent tests were run for all parameter sets, removing the lowest and highest runtimes (5–5%) and computing the average of the remaining values.

TE2 was an Ubuntu Linux based Power System S822LC node in the IBM's Accelerator Lab for High Performance Computing cluster. This is a two socket system equipped with the followings: 2 POWER8 10 Core CPUs, up to 1TB System Memory, and 4 NVIDIA P100 GPUs with NVLink connection. This system includes the exclusive NVLink high bandwidth interconnect between the POWER8 CPUs and NVIDIA GPUs, providing unprecedented bandwith between the CPU and GPUs not available on any other platform. Each P100 GPU is connected to one of the CPUs with 80 GB/sec bi-directional bandwith NVLink connections, and each pair of P100 GPUs are also directly connected to each other with 80GB/sec bi-directional bandwidth NVLink connections (Fig. 3). The same codebase was compiled with gcc and CUDA 8.0 tools using the *arch=compute_60,code=sm_60* flags. The benchmarking methodology was the same as for TE1.

Several tests were run using different population sizes ($P$), where $P = 10, 20, 30 \ldots 2,000$. As a second parameter, the number of GPU devices ($G$) was changed, too. $G = 1, 2$ for the

**Table 1  Runtime values for different population sizes with the GeForce Titan Black cards.** Each column shows the elapsed time for one step of the entire process (memory copy from host to device; kernel execution; memory copy from device to host) in the case of single and dual GPU configurations. The last column shows the speed up total time.

| Size | H → D (µs) | | Kernel (µs) | | D → H (µs) | | Speed-up |
|------|------|------|------|------|------|------|------|
| | 1 GPU | 2 GPUs | 1 GPU | 2 GPUs | 1 GPU | 2 GPUs | $\frac{\sum 1GPU}{\sum 2GPUs}$ |
| 10 | 985 | 1,233 | 146,294 | 121,549 | 1,306 | 2,107 | 1.19 |
| 70 | 1,025 | 1,131 | 375,013 | 144,549 | 1,319 | 1,379 | 2.57 |
| 150 | 1261 | 1,071 | 571,440 | 316,126 | 1,321 | 1,811 | 1.80 |
| 200 | 1,389 | 1,101 | 755,850 | 386,924 | 1,349 | 1,874 | 1.95 |
| 300 | 1,706 | 1,160 | 975,547 | 569,181 | 1,294 | 1,892 | 1.71 |
| 400 | 1,857 | 1,156 | 1,338,839 | 755,804 | 1,289 | 1,861 | 1.77 |
| 500 | 2,139 | 1,258 | 1,701,178 | 931,072 | 1,281 | 1,854 | 1.82 |
| 1,000 | 3,251 | 1,829 | 3,256,236 | 1,697,402 | 1,301 | 1,906 | 1.92 |
| 1,500 | 4,408 | 2,249 | 4,839,201 | 2,476,768 | 1,307 | 1,894 | 1.95 |
| 2,000 | 5,595 | 2,329 | 6,436,735 | 325,6094 | 1,347 | 1,864 | 1.98 |

**Table 2  Total runtime measured with the P100 cards using different population size and number of GPUs.** The last four column shows the speed-up compared to the dual GeForce Titan Black configuration.

| Size | P100 total runtime (µs) | | | | Speed-up | | | |
|------|------|------|------|------|------|------|------|------|
| | 1 GPU | 2 GPUs | 3 GPUs | 4 GPUs | 1 GPU | 2 GPUs | 3 GPUs | 4 GPUs |
| 10 | 113,169 | 121,021 | 126,504 | 136,640 | 1.1 | 1.03 | 0.99 | 0.91 |
| 60 | 150,220 | 118,646 | 123,225 | 133,152 | 0.9 | 1.14 | 1.1 | 1.01 |
| 100 | 151,407 | 121,210 | 124,574 | 131,500 | 1.08 | 1.35 | 1.31 | 1.25 |
| 150 | 215,939 | 156,227 | 126,208 | 129,586 | 1.48 | 2.04 | 2.53 | 2.46 |
| 200 | 275,368 | 158,652 | 161,757 | 134,157 | 1.42 | 2.46 | 2.41 | 2.91 |
| 250 | 345,573 | 222,044 | 163,408 | 171,787 | 1.58 | 2.46 | 3.34 | 3.18 |
| 350 | 488,600 | 281,892 | 228,032 | 170,615 | 1.2 | 2.08 | 2.57 | 3.44 |
| 450 | 615,683 | 350,498 | 228,424 | 240,026 | 1.25 | 2.2 | 3.38 | 3.21 |
| 500 | 620,517 | 351,773 | 229,557 | 236,969 | 1.51 | 2.66 | 4.07 | 3.94 |
| 700 | 954,014 | 499,165 | 360,445 | 295,467 | 1.22 | 2.32 | 3.22 | 3.93 |
| 1,000 | 1,301,749 | 627,025 | 46,3151 | 370,042 | 1.31 | 2.71 | 3.67 | 4.6 |
| 1,500 | 1,891,230 | 972,337 | 636,643 | 517,577 | 1.31 | 2.55 | 3.9 | 4.79 |
| 2,000 | 2,536,917 | 1,310,073 | 909,350 | 640,883 | 1.29 | 2.49 | 3.59 | 5.09 |

first environment, and $G = 1, 2, 3, 4$ for the second environment. To compare the CPU and GPU performance, the same tests were run using a different number of CPUs ($C$), where $C = 5, 10, 20$. The executed steps of the DHCP process are not affected by the actual HTC values. Therefore, predetermined input values were used for the simulations (instead of random chromosomes from a real genetic algorithm).

## GPU runtimes

Table 1 and Fig. 4 show the GPU results for TE1, and Table 2 and Fig. 5 show the GPU results for TE2.

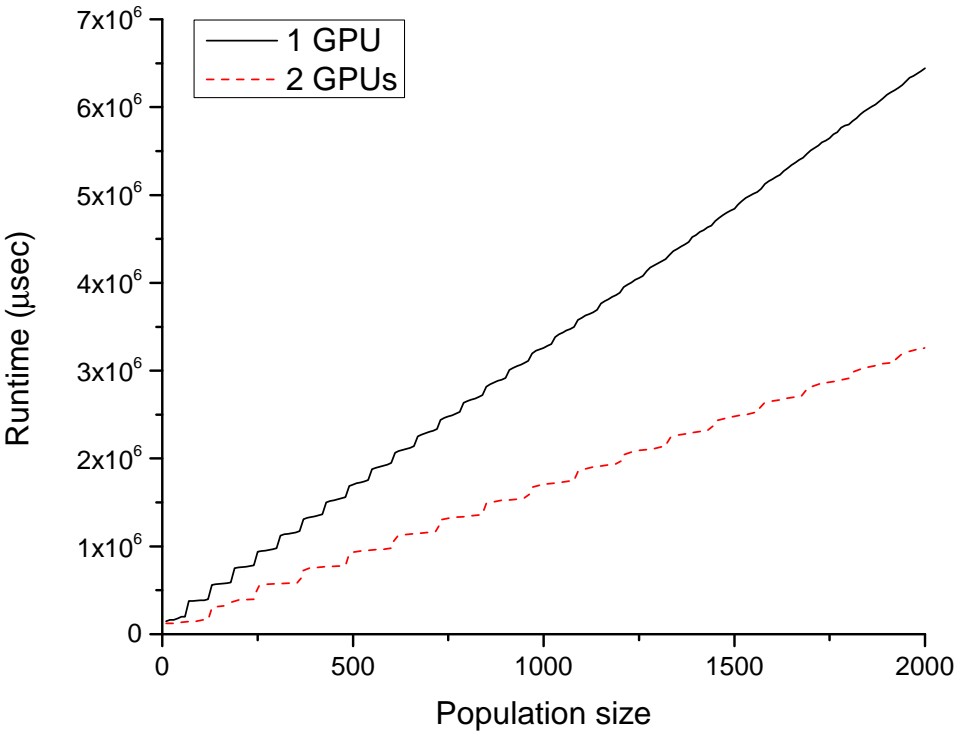

**Figure 4** Runtime (μs) for different population sizes with GeForce Titan Black cards.

As expected, the required runtime inversely linearly correlates with the number of devices. However, in practice, this is a bit more complex, because the series showing the runtimes in both figures are not straight lines, which is caused by the speciality of GPU hardware. This fragmentation is best visible in the case of the one GTX Titan Black card (black solid series in Fig. 4). The total runtime is slightly increased from population sizes 10 to 60, but the required runtime for population size 70 is almost twice as large. The explanation for this phenomenon is based on the GPU architecture. The GTX Titan Black has 15 streaming multiprocessors (each of them containing 192 processing units), and every multiprocessor can execute four heat transfer simulations simultaneously. Accordingly, launching one or 60 parallel simulations takes a similar time. In the case of a greater number of threads, the scheduling (including memory transfers) becomes more complex; consequently, the effect of these steps becomes less sharp.

As visible in Fig. 5, these steps already exist for one P100 card, the runtime increases in every 50–60th step. The P100 has 56 multiprocessors, each of these can execute one simulation at the same time; therefore, the device can run 56 thermal history generations in parallel. This regularity can also be observed in the case of multiple GPUs; for example, in the case of 4 P100 devices, steps from 220 to 230 ($56 \cdot 4 = 226$) and from 440 to 450 ($56 \cdot 4 \cdot 2 = 448$) has a significant impact to the runtime. It is worth evaluating as large a population as possible at the same time. Thus, the recommended population size should be near to these limits from below (220 or 440).

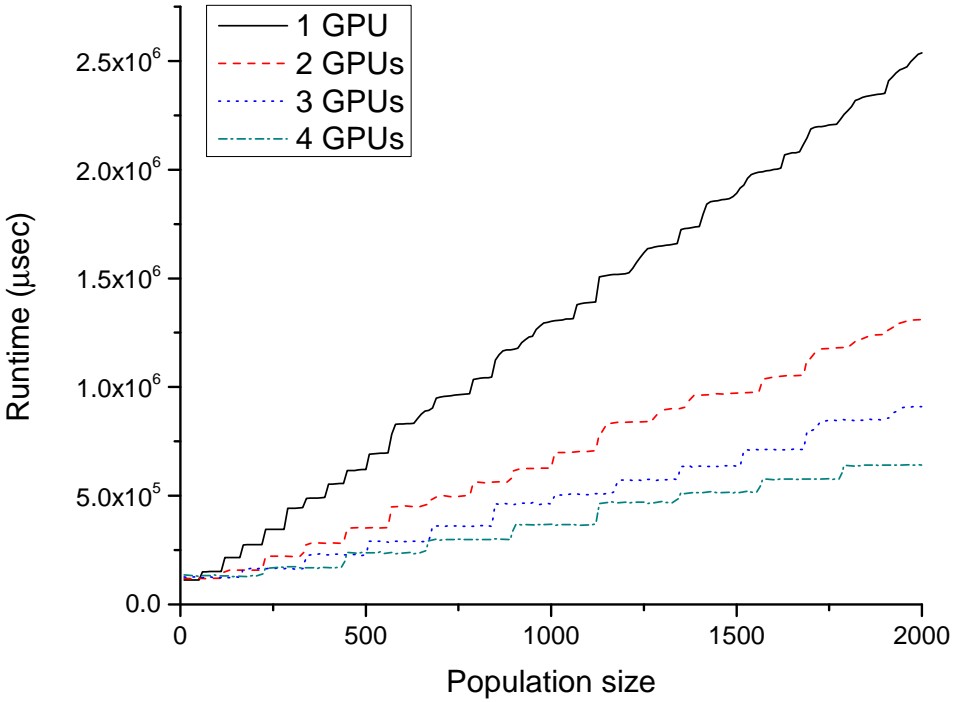

**Figure 5** Runtime (μs) for different population sizes with P100 cards.

**Table 3 Runtime of the DHCP solver for different population sizes and CPU core counts (thread counts).** Last two columns show the measured speed-up for configurations with 20 CPU cores compared to five CPU cores, and four P100 GPUs compared to 20 CPU cores.

| Population size | 5 cores | 20 cores | Speed-up | |
|---|---|---|---|---|
| | (μs) | (μs) | 5C → 20C | 20C → 4G |
| 10 | 1,197,806 | 1,023,642 | 1.17 | 7.49 |
| 100 | 12,233,650 | 5,068,359 | 2.41 | 38.54 |
| 500 | 60,234,800 | 21,247,890 | 2.83 | 89.67 |
| 1,000 | 121,159,900 | 39,240,730 | 3.09 | 106.04 |
| 2,000 | 241,941,200 | 78,851,140 | 3.07 | 123.04 |

## CPU runtimes

Table 3 and Fig. 6 show the CPU results for TE2.

CPU performance analysis is not in the focus of this paper, but these benchmarks have been run only for the CPU–GPU comparison. As visible, the increase in the number of cores effectively increases the performance (each core is responsible for one heat transfer simulation). On the other side, as the population size was increased, the runtime increased almost linearly.

In the case of CPU implementations, the implementation is much simpler. There is no need to transfer input data from the host to the device and the output data from the device to the host, and kernel launch overhead is also missing. Therefore, the expectation was that

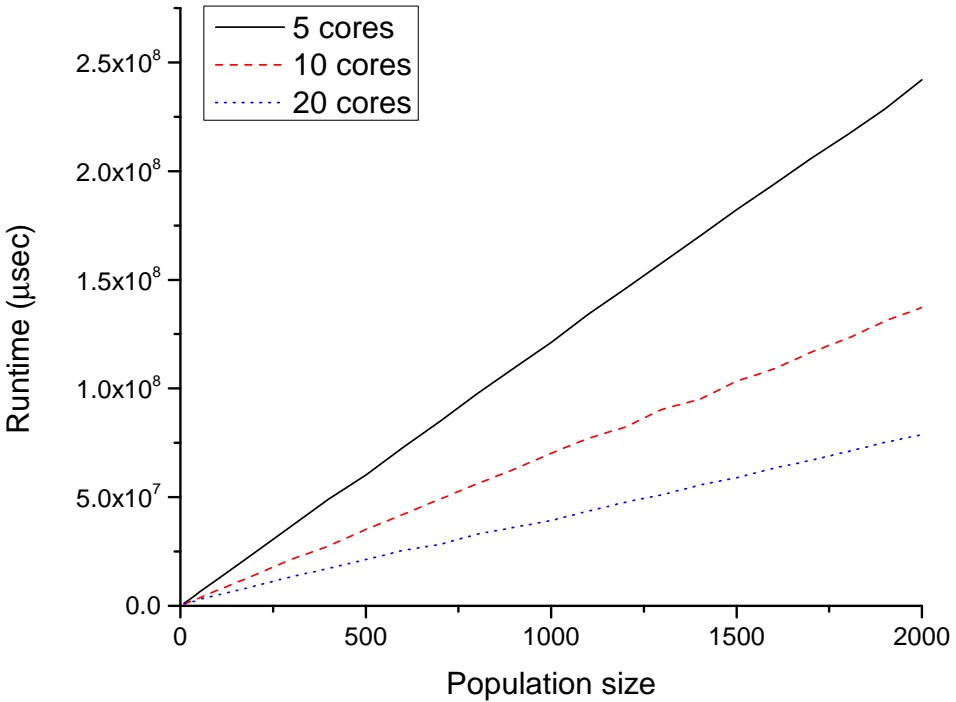

**Figure 6** Runtime (μs) for different population sizes with POWER CPUs.

the CPU would be faster in the case of small population sizes (where the GPUs cannot take advantage of the high number of multiprocessors), but as is visible, this was not true.

Comparing these results to the GPU runtimes, it is evident that it is not worth using the CPU for any population size. In the case of 20 parallel heat transfer simulations (which is the ideal configuration for the server with 20 CPU cores), all P100 GPU implementations were 8–9 times faster. Above this population size, the difference becomes even bigger. In the case of 2,000 parallel simulations, the P100 cards were 123 times faster.

It also raises the question of whether it is worth implementing and using a hybrid solution combining the CPUs and GPUs together. For small population sizes, the answer is no. For example, in the case of 200 chromosomes, if the GPUs evaluate 199 of them (needs about 134 ms) and the CPUs evaluate only one (needs about 1,024 ms), the overall runtime will be higher (max(134 ms, 1,024 ms) = 1,024 ms) than the plain GPU implementation. The only occasion when it is worth considering the hybrid implementation is a large population size near the previously explained runtime steps from above. For example, in the case of 1,690 chromosomes and two GPUs, it would be worth to assign 20 of them to the CPUs. The CPU runtime will be about 1,024 ms, the GPU runtime for the remaining is 1,054 ms. Consequently, the overall execution time is 1,054 ms. Using the GPU implementation exclusively, it requires more, 1,132 ms. In the case of four GPUs, it would also be possible to find a similar situation, but not in the examined 10–2,000 population size interval.

**Table 4 Details of the Host to Device memory transfer for the GeForce Titan Black cards.** The first column shows the size of the population; the second one contains the total required memory for the chromosome data. The following columns contain the runtime of device memory allocation and data transfer. Data transfer rate (DTR) is the amount of data that is moved in a given time.

| Size | Data (byte) | Allocation ($\mu$s) | | Copy ($\mu$s) | | DTR ( GB/s) | |
|---|---|---|---|---|---|---|---|
| | | 1 GPU | 2 GPUs | 1 GPUs | 2 GPUs | 1 GPU | 2 GPUs |
| 10 | 13,600 | 817 | 930 | 168 | 304 | 0.08 | 0.04 |
| 50 | 68,000 | 795 | 863 | 251 | 309 | 0.25 | 0.21 |
| 100 | 136,000 | 812 | 883 | 368 | 286 | 0.34 | 0.44 |
| 200 | 272,000 | 777 | 806 | 611 | 295 | 0.41 | 0.86 |
| 300 | 408,000 | 784 | 826 | 922 | 334 | 0.41 | 1.14 |
| 400 | 544,000 | 784 | 801 | 1,073 | 355 | 0.47 | 1.43 |
| 500 | 680,000 | 891 | 841 | 1,249 | 417 | 0.51 | 1.52 |
| 1,000 | 1,360,000 | 856 | 1,156 | 2,395 | 673 | 0.53 | 1.88 |
| 1,500 | 2,040,000 | 905 | 1,336 | 3,504 | 913 | 0.54 | 2.08 |
| 2,000 | 2,720,000 | 982 | 1,183 | 4,613 | 1,146 | 0.55 | 2.21 |

**Table 5 Details of the Host to Device memory transfer for the P100 cards.** The first column shows the size of the population; the second one contains the total required memory for the chromosome data. The following columns contain the runtime of device memory allocation and data transfer. Data transfer rate (DTR) is the amount of data that is moved in a given time.

| Size | Data (byte) | Allocation ($\mu$s) | | Copy ($\mu$s) | | DTR (GB/s) | |
|---|---|---|---|---|---|---|---|
| | | 1 GPU | 2 GPUs | 1 GPUs | 2 GPUs | 1 GPU | 2 GPUs |
| 10 | 13,600 | 648 | 1,806 | 14 | 18 | 0.94 | 0.7 |
| 50 | 68,000 | 644 | 1,810 | 15 | 19 | 4.21 | 3.28 |
| 100 | 136,000 | 646 | 1,788 | 17 | 20 | 7.4 | 6.28 |
| 200 | 272,000 | 647 | 1,790 | 22 | 23 | 11.72 | 11.2 |
| 300 | 408,000 | 645 | 1,789 | 26 | 26 | 14.65 | 14.84 |
| 400 | 544,000 | 647 | 1,804 | 30 | 29 | 16.92 | 17.61 |
| 500 | 680,000 | 646 | 1,802 | 34 | 32 | 18.54 | 19.72 |
| 1,000 | 1,360,000 | 647 | 1,802 | 55 | 49 | 23.03 | 25.88 |
| 1,500 | 2,040,000 | 653 | 1,789 | 77 | 67 | 24.64 | 28.55 |
| 2,000 | 2,720,000 | 654 | 1,794 | 96 | 80 | 26.3 | 31.78 |

## Data transfer rates

During the tests, the data transfer rates were also recorded in the case of TE1 and TE2. Tables 4–5 and Figs. 7–8 show these results. The advance of the new NVLink based Power CPU–GPU connection is the high transfer bandwidth between the CPU and the GPU memory. As visible from the results, this works well in practice.

Table 4 shows that the measured results for TE1 were far from the theoretical maximal transfer speed and it is also visible, that using two GPUs have advantages compared to the single GPU configuration. These devices are installed into different PCI-E slots, so their data transfers can be processed in parallel (both cards were installed into PCI-E 16× slots, but one of them works at 4× speed, and the other one at 8× speed). In the case

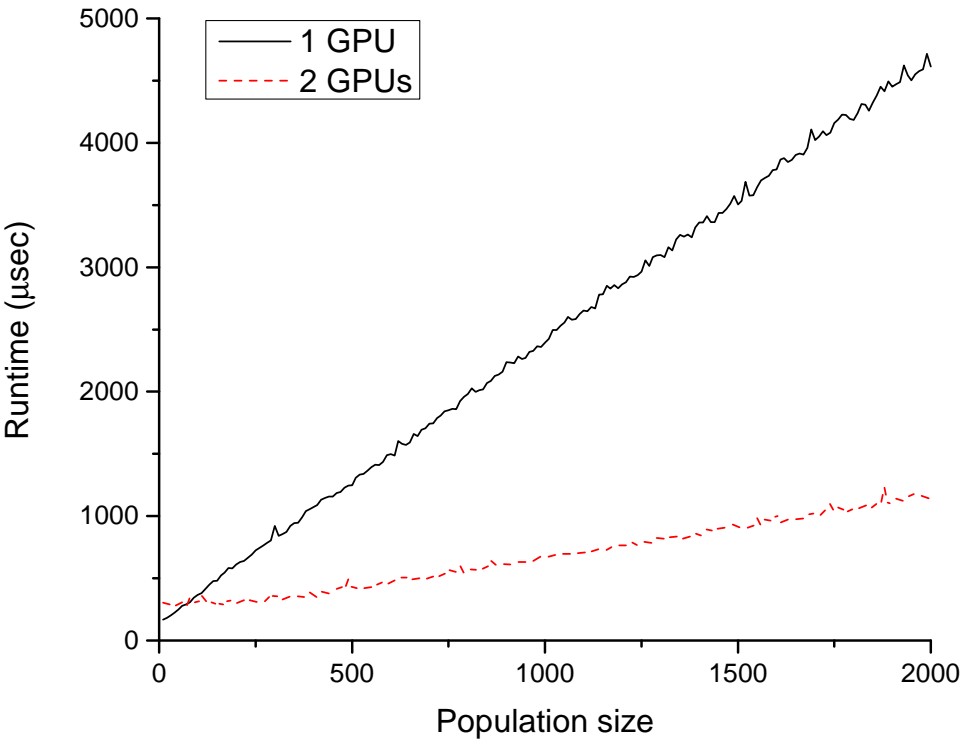

**Figure 7** **Memory transfer time (μs) for different population sizes with GeForce Titan Black cards.**

of the CPU–GPU NVLink architecture (Table 5), the bandwidth was significantly higher (however, the amount of data is still low to achieve the theoretical maximum).

It is worth noting that with the two GeForce GPUs, the memory transfer time is well over twice as fast as with one card; in sharp contrast, for two P100 cards it is only about 40% faster than that with one card. And what is more interesting, the usage of three or four devices does not have any benificial effect. The reason for this is that the used P100 node has a special architecture in that one CPU socket and a maximum of two GPUs form an "island" in which any pair of the triad has fast data transfer. In contrast, transfer of data between islands is slower. In the case of $N_G$ number of GPUs, the IHCP solver code uses $N_G$ individual CPU threads to manage the data transfers between the host and the devices. This practice was satisfactory for TE1, but leads to the experienced strange behaviour for TE2, because all of these threads were scheduled on the same physical processor. To reach the maximum transfer rate with four devices, multiple CPU threads should be launched and bounded to logical CPUs that are appropriately placed for the GPUs we intend to use. However, customizing the already existing codebase to a specified architecture is out of the scope of this paper, especially in view of the fact that the NVLink capable server was even significantly faster without any fine-tuning.

Comparing the GTX and P100 based configurations, the latter was 12–49 times faster. The new NVLink architecture is, therefore, faster, and its overall transfer rate becomes higher with multiple cards. This speed up is achievable in the case of small population sizes

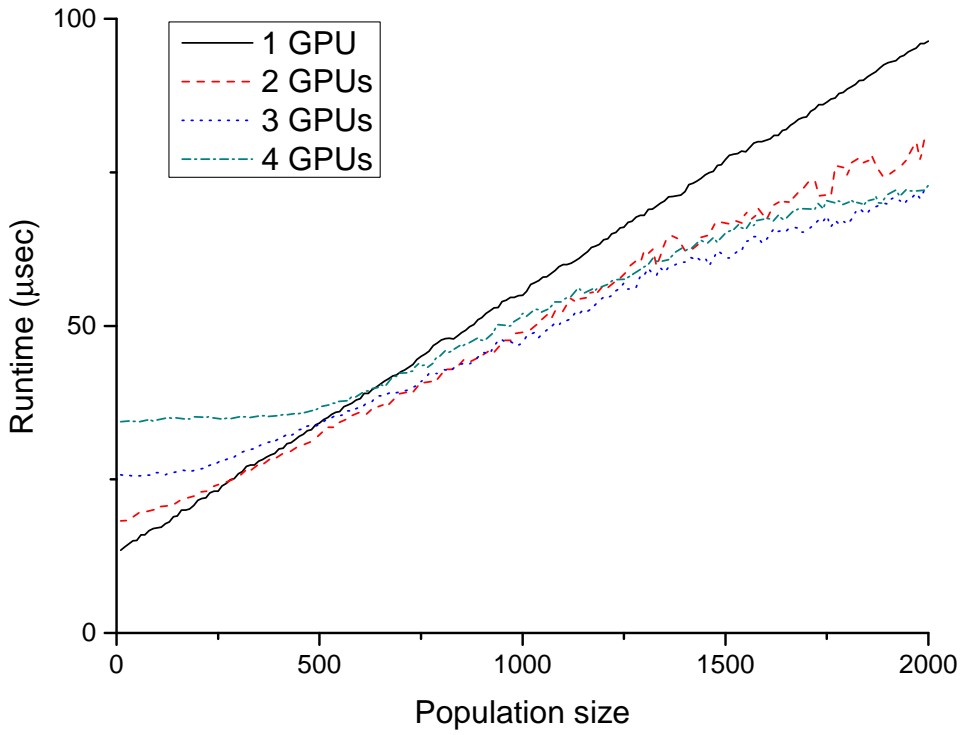

**Figure 8** **Memory transfer time (µs) for different population sizes with P100 cards.**

(it is 16 times quicker in the case of 20 chromosomes and 2 GPUs). Therefore, this novel architecture is highly recommended for running the IHCP solver GAs.

## CONCLUSIONS

A novel data-parallel algorithm to solve the IHCP and a GPU based implementation was outlined. By using a higher level of parallelism, it can use the processing power of current multi-GPU systems. Analysing the architecture of the new P100 based servers and the runtime results, the conclusions are as follows:

- In the case of the IHCP, the runtime of both the CPU and the GPU implementations is nearly linearly depending on the population size, and inversely on the number of processing cores. In the case of GPUs, the number of devices and the multiprocessor architecture makes this correlation more complex, the runtime increasing significantly at given (predictable) population sizes.
- According to this observation, the recommended population size is close to these points from below for the exclusive GPU implementations. In the case of hybrid systems, the most efficient population size should be close to these points from above.
- The NVLink connection between the CPU and GPUs can significantly decrease the data transfer time. It is also faster for small population sizes; however, the maximal bandwidth is not achievable in these cases. The results of this work encourage the use of multiple

graphics accelerators for the purposes of heat transfer simulations. To fully utilise the processing power of all GPUs, it is necessary to reach a higher level of parallelism, but there are several subfields (like the IHCP) where it is feasible.

As future work, it is worth fine tuning the algorithm to the new IBM Power System architecture. Only a naïve porting of the already existing CUDA algorithms have been used to this new environment without any major changes. It deserves a deeper study to see why only one block is scheduled into each multiprocessor of the P100 devices. If it is possible to use some minor configuration changes (decreasing the number of registers or shared memory) to run multiple blocks, then this can double the performance of the system.

### Funding

The author received financial support from the Hungarian State and the European Union under the EFOP-3.6.1-16-2016-00010 project. NVIDIA Corporation provided graphics hardware for the GPU benchmarks through the CUDA Teaching Center program. This work was additionally supported by the ÚNKP-17-4/I New National Excellence Program of the Ministry of Human Capacities. The funders had no role in study design, data collection and analysis, decision to publish, or preparation of the manuscript.

### Grant Disclosures

The following grant information was disclosed by the author:
Hungarian State and the European Union.
NVIDIA Corporation.
ÚNKP-17-4/I New National Excellence Program of the Ministry of Human Capacities.

### Competing Interests

Sándor Szénási is an Academic Editor for PeerJ Computer Science.

### Author Contributions

- Sándor Szénási conceived and designed the experiments, performed the experiments, analyzed the data, contributed reagents/materials/analysis tools, wrote the paper, prepared figures and/or tables, performed the computation work, reviewed drafts of the paper.

### Data Availability

All raw benchmark data have been uploaded as Supplemental Files.

### Supplemental Information

Supplemental information for this article can be found online at http://dx.doi.org/10.7717/peerj-cs.138#supplemental-information.

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
