# Peer review of "Solving the inverse heat conduction problem using NVLink capable Power architecture"

_PeerJ Computer Science, doi:10.7717/peerj-cs.138_

## Round 0.1 · original submission · Major Revisions

Please address the reviewers comments when preparing your revision.

Reviewer 1 ·

Basic reporting

The authors have presented a solution to inverse heat conduction problem, IHCP, for cylindrical geometry with the help of GPU and CUDA. They attain a tremendous speed-up of 120 over with 4 GPUs and 20 cores.
Although the authors call it a 2-D problem, in essence it is a 3-D problem with axial symmetry.
The direct heat conduction problem, whose temperatures are used to estimate heat transfer coefficients could be more completely described for understanding rather than incomplete description ( boundary values at z=0, z=L , missing) given in the paper.

The authors are, however, commended for achieving such a high speed-up with just 4 GPUs, under CUDA environment and for achieving high parallelism to solve this problem.

The paper is well written, graphs are clear, the background has been sufficiently provided.

Experimental design

The research problem, IHCP, although well defined, could be described mathematically correctly by stating the region of equation and boundary conditions as mentioned in another section of the review.

The massive parallelism achieved through shared memory architecture is well described.

Bench marking is very well done with various GPUs and a CPU and is described as well as depicted through graphs and text in detail.

It would be better if authors could provide more explanations on the experimental setup and data results. For example, the configurations of the hardware and the details on how the simulations are run.

Validity of the findings

Even the DHCP are are not much attempted in the literature, with massively parallelism achieved, through GPUs and CUDA environment, the authors attempting an inverse heat conduction problem with GPU and CUDA is pioneering.

Conclusion are well stated and clear, and amazing results are obtained with the usage of small number (4) of GPUs with larger of cores size (20).

Additional comments

The authors have attempted a very useful and pioneering work in this paper. They have achieved a tremendously high speed-up of 120 with just 4 GPUs under CUDA environment. The authors are commended this work.
The description of the boundary conditions although clear at the circular boundary r=R, they are not correctly stated at the boundaries z=0, z=L, a region is cited instead of the boundaries. The authors are recommended to state the region (0<z<L, 0<r<R) for equation number 1 as well in addition to correcting the boundary values at z=0, z=L.

Reviewer 2 ·

Basic reporting

-To put all measured data of the experiments in the paper is redundant, there should be a good design on how to present data efficiently

- All tables (1,2,3,4 and 5 ) include huge amount of numbers, it is not necessary to put all the population sizes in the paper, some important points of the population size can be picked to show the trend of changing execution time or memory transfer time.
Also what is the difference between showing using 1 or 2 or 3 or 4 GPUs or 5-10-20 cores, it is only for the purpose of showing the trend, you can pick only 2 sizes

-Table 1 and table 3 would be good to see the speed up factor, in table 3 speed up factor comparing CPU vs. GPU is important

Experimental design

-Why two sets of multi GPUs with two architectural set up are used? (Geforce, P100) where do you use it in your analysis or does that help to draw any conclusion based on these two architectural configurations? for example you can compare 2 GPUs from one configuration to the another

-The memory Transfer time in Fig 6 and Fig 7 in two different GPU configurations is different, why any explanation or analysis?

-The NVLink experimental part needs more explanation and analysis.

Validity of the findings

no comment

---

## Round 0.2 · Minor Revisions

The reviewers felt that you addressed the major concerns from the previous version of your paper, but would still like to see some remaining minor revisions before it is published.

Reviewer 1 ·

Basic reporting

The paper is well written in clear and concise form. The Problem of Inverse Heat conduction for estimating the heat transfer coefficients, which are many times not accurately measured or available., is very well described mathematically, and solved for a finite solid cylinder.

The good background is provided as to what other attempts have been made to determine the coefficients by other methods and why the author chose this technique to determine or estimate the coefficients through this GPU technology

The results are clearly graphed and discussed.

Experimental design

Experimental design has been clearly stated and implemented.

However algorithm description is missing. It would be helpful to understand the experimental design if the authors could add algorithm description.

Validity of the findings

Results are very well depicted through graphs, and performance comparisons have been made with different core sizes (5, 10, 20) as well as through various number of GPUs (1,2,3,4). Good results and high performances have been achieved (120 Times).

Additional comments

Well written paper. Tackling of inverse heat conduction to estimate various heat transfer coefficients is difficult but adeptly solved. The authors deserve credits for handling a time consuming and difficult problem through GPU technology. The three dimensional problem is described as 2-dimensional problem through axial cylindrical geometry. It is remarkably handled, graphed, comparative performance depicted through various GPUs/CPUs.

Reviewer 2 ·

Basic reporting

Comparing to last version the representation of the experimental data improved and less redundant experimental data is in the paper. Also speed up comparison is added.
Please edit Table 1 caption in the manuscript the last sentence should be :
the last column shows the speed up total time.

Experimental design

It is good that more detailed mathematical background for the problem is added to paper, this helps to understand the inverse heat conduction problem better.
Data transfer rate section is convincingly reconstructed and explanation added that improves the result discussion on that part.

Validity of the findings

Paper shows promising speed-up results using multi-GPU in inverse heat conduction.

---

## Round 0.3 · accepted · Accept

Thank you for addressing the issues raised by the reviewers.